# Anomalous Behavior of the Non-Hermitian Topological System with an Asymmetric Coupling Impurity

**DOI:** 10.3390/e27010078

**Published:** 2025-01-17

**Authors:** Junjie Wang, Fude Li, Weijun Cheng

**Affiliations:** 1College of Digital Technology and Engineering, Ningbo University of Finance & Economics, Ningbo 315175, China; 2College of Computer and Information Engineering, Tianjin Agricultural University, Tianjin 300384, China; lifd283@nenu.edu.cn; 3School of Integrated Circuits, Tsinghua University, Beijing 100084, China; chengwj177@nenu.edu.cn

**Keywords:** impurity, non-Hermitian system, quantum phase transition

## Abstract

A notable feature of systems with non-Hermitian skin effects is the sensitivity to boundary conditions. In this work, we introduce one type of boundary condition provided by a coupling impurity. We consider a system where a two-level system as an impurity couples to a nonreciprocal Su–Schrieffer–Heeger chain under periodic boundary conditions at two points with asymmetric couplings. We first study the spectrum of the system and find that asymmetric couplings lead to topological phase transitions. Meanwhile, a striking feature is that the coupling impurity can act as an effective boundary, and asymmetric couplings can also induce a flexibly adjusted zero mode. It is localized at one of the two effective boundaries or both of them by tuning coupling strengths. Moreover, we uncover three types of localization behaviors of eigenstates for this non-Hermitian impurity system with on-site disorder. These results corroborate the potential for control of a class of non-Hermitian systems with coupling impurities.

## 1. Introduction

In recent years, non-Hermitian physics has attracted a plethora of attention, uncovering a wide range of phenomena and applications in both classical and quantum systems [1,2,3,4,5,6,7,8,9,10,11,12,13,14,15,16,17,18,19,20,21,22,23,24,25,26,27,28,29,30,31,32,33,34,35,36,37]. Non-Hermitian systems also exhibit some markedly different properties with no conventional Hermitian counterparts, such as biorthogonal eigenstates [2], exceptional points [5,6], and the breakdown of the conventional bulk–boundary correspondence [15,16,17,18].

Another unique feature of the non-Hermitian system is the accumulation of all eigenstates at the boundaries, which is a phenomenon dubbed the “non-Hermitian skin effect” [17]. A notable feature of systems with non-Hermitian skin effects is that the properties of both spectrum and eigenstates may be dramatically changed by turning the boundary conditions from periodic to open ones. In between, an impurity introduced into the system could also play the role of the boundary [38,39,40,41,42,43,44,45]. A striking feature of the impurity model is that boundary impurities can generate new types of steady-state localization behavior characterized by scale-free accumulation of eigenstates [38]. Due to the fact that the energy shift of the system can be extraordinarily changed by adding a vanishingly small boundary impurity, this kind of system can also be harnessed to devise sensors in an experimentally realistic setting [46,47,48,49].

Recently, simulations of topological systems using superconducting quantum circuits have attracted a great deal of attention [50,51,52,53]. On the other hand, superconducting qubits acting as giant artificial atoms have played an important role in superconducting quantum circuits. They can be nonlocally coupled to a waveguide at multiple points [54,55,56,57,58,59]. It was observed that the giant atom can act as an effective boundary and induce chiral zero modes for the waveguide in Hermitian topological systems [58,59].

These studies also further stimulate a new research direction for the interaction between the non-Hermitian topological system and quantum emitters [60,61,62,63,64]. In Ref. [63], the researchers found that giant emitters can exhibit essentially different dynamical behaviors by turning the relative strengths of the nonlocal couplings, and a series of unconventional quantum optical phenomena have been unveiled, such as nonreciprocal decoherence-free interaction. In Ref. [64], focusing on spectrum structures and the localization of eigenstates for the system that a giant atom as an impurity couples to a non-Hermitian topological chain with the same nonlocal coupling strengths, the authors mainly found that the impurity can induce asymmetric zero modes. This begs the question that what new physical phenomena will emerge in this type of systems by leveraging the relative strengths of the nonlocal couplings.

In this work, we focus on a system composed of a two-level system as an impurity and a nonreciprocal Su–Schrieffer–Heeger (SSH) chain with asymmetric couplings. We first study the fundamental properties of the spectrum and find that asymmetric couplings can cause topological phase transitions in an A−B coupling case. As two coupling strengths $g_{n}$ and $g_{m}$ become more and more different, it seems that the zero mode will always exist. We further reveal the localization behaviors of zero mode for the system. It can be localized at one of the two effective boundaries or both of them depending on the relative strengths of the nonlocal couplings. We also show three types of localization behaviors of all eigenstates for systems with on-site disorder in the end.

The paper is organized as follows. In Section 2, we introduce a model to describe a system composed of a two-level system and a nonreciprocal SSH chain. In Section 3, firstly, we show the spectrum of the system, and we give the reason for the occurrence of topological phase transitions. Secondly, we derive analytical expressions and show numerical simulations for the zero modes. In the end, we introduce the mean center of mass (mcom) to describe the localization feature of all eigenstates with on-site disorder. In Section 4, we summarize our results.

## 2. Model and Methods

We consider a nonreciprocal SSH chain with the periodic boundary conditions (PBCs) in real space. The non-Hermitian Hamiltonian associated with this chain can be written as follows:(1)$$\begin{array}{cc} H_{\mathrm{SSH}}= & \sum_{l=1}^{L}[\left(t_{1}+\gamma \right){\hat{C}}_{A,l}^{†}{\hat{C}}_{B,l}+\left(t_{1}-\gamma \right){\hat{C}}_{B,l}^{†}{\hat{C}}_{A,l}\\ \; & +t_{2}{\hat{C}}_{A,l+1}^{†}{\hat{C}}_{B,l}+t_{2}{\hat{C}}_{B,l}^{†}{\hat{C}}_{A,l+1}], \end{array}$$
where the chain is composed of *L* unit cells, with each containing two sites. ${\hat{C}}_{A(B),l}^{†}$ and ${\hat{C}}_{A(B),l}$ are the creation and annihilation operators for the sublattice site A(B) at the *l*-th unit cell. The parameters $t_{1}\pm \gamma$ and $t_{2}$ are intracell and intercell couplings. The asymmetry of hopping amplitudes (γ≠0) leads to the non-Hermiticity of the system.

We here focus on analyzing what occurs when a two-level impurity couples to a nonreciprocal SSH chain with asymmetric coupling, as schematically shown in Figure 1. Hence, we introduce a two-level impurity coupling at two points to a nonreciprocal SSH chain via A−B couplings [Figure 1a] or A−A couplings [Figure 1b], where nonlocal coupling points locate at *n*-th lattice site and *m*-th lattice site. The system with B−B couplings and system with A−A couplings are very similar, so we do not study the case with B−B couplings. Without loss of generality, we hereafter assume n<m. The interaction Hamiltonian between the impurity and the nonreciprocal SSH chain is given as(2)$$\begin{array}{cc} H_{I,AB} & =g_{n}\sigma^{+}{\hat{C}}_{A,n}+g_{m}\sigma^{+}{\hat{C}}_{B,m}+H.c.,\\ H_{I,AA} & =g_{n}\sigma^{+}{\hat{C}}_{A,n}+g_{m}\sigma^{+}{\hat{C}}_{A,m}+H.c., \end{array}$$
where $g_{n}$($g_{m}$) is the coupling strength between the impurity and the *n*-th (*m*-th) site of the nonreciprocal SSH chain. $\sigma^{+}=\left|e\rangle \langle g\right|$ is the usual pseudospin ladder operator, and |g〉 and |e〉 are the ground state and the excited state of the impurity, respectively. The total Hamiltonians of the atom–chain coupling can be expressed as(3a)$$H_{AB}=H_{\mathrm{SSH}}+H_{I,AB},$$(3b)$$H_{AA}=H_{\mathrm{SSH}}+H_{I,AA}.$$
We have assumed that the impurity is resonant with the energy band center, i.e., frequency of impurity is zero. For the particularly experimental scheme, adding a constant imaginary shift to all sites corresponding to a passive setting with loss only [65], this correction does not affect the localization of eigenstates or the existence of boundary modes.

This model can be observed in a range of experimental settings, including electrical circuits [33,49] and photonic systems [46]. For example, there are 2L nodes in our designed non-Hermitian topological circuit. The intercell coupling of the circuit is fulfilled by a capacitor. The nonreciprocal coupling can be achieved through connecting capacitors in series with a voltage follower. Due to the virtual open and virtual short circuit conditions between the inverting input and noninverting input pins, the current at the one side of the capacitor is blocked, while it remains uninfluenced at the other side. Impurity can be realized by using Josephson junctions [66] or just a node in the circuit [49]. The coupling between the impurity and chain is achieved by connecting a capacitor. And then the variation of coupling strength can be achieved by adjusting the capacitance. In addition, this model can also be thought of as an array of coupled optical ring resonators [46]. The asymmetric coupling has been experimentally achieved by introducing two scatterers into the mode volume of a ring resonator. The intercell coupling may be achieved by chiral couplers. The coupling between impurity and chain may be represented by any optically impenetrable region imposing a tunneling barrier. The variation of coupling strength then only needs to change the refractive index of the impenetrable medium.

## 3. Results

### 3.1. Spectrum and Topological Phase Transition

To illustrate the role of coupling impurity in the SSH chain. In Figure 2a, we first show the spectrum of pure SSH chain under periodic boundary conditions in the complex plane as a contrast. We also show spectrum of the system with A−B couplings ([Disp-formula FD3a-entropy-27-00078]) and A−A couplings ([Disp-formula FD3b-entropy-27-00078]) in Figure 2b,c, respectively. Note that an obvious feature is that the impurity can induce zero modes. This implies that the impurity can act as an effective boundary for a nonreciprocal SSH chain. Except for the zero modes, what are also called middle bound states, there are other eigenvalues outside the continuous bands. According to the value of the real part of the spectrum as a comparison, we call the corresponding eigenstates upper and lower bound states, respectively. The feature of these bound states will be described in the next section. Then, we study the spectrum feature of systems with A−B couplings and A−A couplings in detail.

Firstly, for the A−B coupling case ([Disp-formula FD3a-entropy-27-00078]), we show the absolute value of the spectrum as a function of $t_{1}$ in Figure 3. The results were obtained by numerically solving the Schödinger equation. The asymmetric coupling strengths were set as $g_{m}=1$, $g_{n}=0.5$, 0.1, 0.01, and 0.001 for (a), (b), (c), and (d), respectively. In fact, the spectrum feature for the system with the same coupling strengths ($g_{n}=g_{m}=1$) has been studied in Ref. [64] in detail. A main finding is the condition for the emergence of the zero mode, i.e., $t_{1}\in \left[-t_{2}+\gamma ,t_{2}-\gamma \right]$. In Figure 3a, one can see that the condition for the emergence of the zero mode in this case ($g_{n}=0.5$, $g_{m}=1$) is identical to the result with the same coupling strengths ($g_{n}=g_{m}=1$). However, this condition no longer holds as the two coupling strengths $g_{n}$ and $g_{m}$ become more and more different, as shown in Figure 3b–d. A striking feature is that when the coupling strength $g_{n}=0.001$ is much smaller than $g_{m}=1$, it seems that zero mode will always exist and not change with the parameters $t_{1}$, as shown in Figure 3d. This implies that asymmetric couplings can cause topological phase transitions.

In order to derive a condition for the emergence of the zero mode and show why asymmetric couplings cause topological phase transitions, we first give the Hamiltonian of the system in the momentum space via Fourier transformation described by(4)$$\begin{array}{cc} H_{AB}\left(k\right)= & H_{\mathrm{SSH}}\left(k\right)+H_{I,AB}\left(k\right)\\ = & \sum_{k}\left[\left[\left(t_{1}+\gamma \right)+t_{2}e^{-ik}\right]{\hat{C}}_{A,k}^{†}{\hat{C}}_{B,k}+\left[\left(t_{1}-\gamma \right)+t_{2}e^{ik}\right]{\hat{C}}_{B,k}^{†}{\hat{C}}_{A,k}\right]\\ \; & +\frac{1}{\sqrt{L}}\sum_{k}\left[\sigma^{+}\left(g_{n}{\hat{C}}_{A,k}e^{ikn}+g_{m}{\hat{C}}_{B,k}e^{ikm}\right)+\;H.c.\;\right], \end{array}$$
where ${\hat{C}}_{A(B),k}^{†}$ and ${\hat{C}}_{A(B),k}$ denote the creation and annihilation operators for the sublattice site A(B) at the *k*-th unit cell in the momentum space. In the single-excitation subspace, the eigenstates of the Hamiltonian $H_{AB}\left(k\right)$ can be written as follows:(5)$$\begin{array}{c} |\Psi \rangle =U_{e}|e,G\rangle +\sum_{k}\alpha_{k}{\hat{C}}_{A,k}^{†}|g,G\rangle +\sum_{k}\beta_{k}{\hat{C}}_{B,k}^{†}|g,G\rangle , \end{array}$$
where |G〉 is the ground state of the SSH chain (vacuum state), and |e(g),G〉=|e(g)〉|G〉 can be used to form a complete base for the whole system. $\alpha_{k}\left(\beta_{k}\right)$ denotes the amplitude for the sublattice site A(B) at the *k*-th unit cell, and $U_{e}$ denotes the amplitude at site of the impurity. With the time-independent Schrödinger equation $H_{AB}\left(k\right)|\psi \rangle =E|\psi \rangle$, the transcendental equation for the energy *E* can be expressed as (See Appendix A for analytical results)(6)$$\begin{array}{cc} E= & \frac{\left(g_{n}^{2}+g_{m}^{2}\right)}{L}\sum_{k}\frac{E}{E^{2}-\omega_{k}^{2}}+\frac{2g_{n}g_{m}}{L}\sum_{k}(t_{1}\cos\left[k\left(m-n\right)\right]\\ \; & -\gamma \sin\left[k\left(m-n\right)\right]+t_{2}\cos\left[k\left(m-n+1\right)\right])/\left(E^{2}-\omega_{k}^{2}\right) \end{array}$$
with(7)$$\omega_{k}=\sqrt{\left(t_{1}+\gamma +t_{2}e^{-ik}\right)\left(t_{1}-\gamma +t_{2}e^{ik}\right)}.$$
Setting E=0, we can obtain a condition for the emergence of the zero mode, i.e., $t_{1}\in \left[-t_{2}+\gamma ,t_{2}-\gamma \right]$. (See Appendix A for analytical results and Figure 3a for numerical simulations). This confirms that the condition for the emergence of the zero mode with asymmetric couplings is identical to the result with the same coupling strengths. However, as $g_{n}\to 0$ ($g_{m}\to 0$ is similar), Equation (Equation 6) can be simplified as(8)$$\begin{array}{cc} E= & \frac{g_{m}^{2}}{L}\sum_{k}\frac{E}{E^{2}-\omega_{k}^{2}}. \end{array}$$
Evidently, E=0 is always the solution of the Equation (Equation 8), which implies that there is always a zero mode in the systems. Hence, the topological phase transitions occur as two coupling strengths $g_{n}$ and $g_{m}$ become more and more different (see Figure 3a–d for numerical simulations). This can also be understood in terms of symmetry. The matrix form of the Hamiltonian $H_{AB}\left(k\right)$ in the local site basis satisfies(9)$$H_{AB}\left(k\right)=\left(\begin{array}{cccccc} 0 & t_{1}+\gamma +t_{2}e^{-ik_{1}} & 0 & 0 & \ldots & \frac{g_{n}e^{-ik_{1}n}}{\sqrt{L}}\\ t_{1}-\gamma +t_{2}e^{ik_{1}} & 0 & 0 & 0 & \ldots & \frac{g_{m}e^{-ik_{1}m}}{\sqrt{L}}\\ 0 & 0 & 0 & t_{1}+\gamma +t_{2}e^{-ik_{2}} & \ldots & \frac{g_{n}e^{-ik_{2}n}}{\sqrt{L}}\\ 0 & 0 & t_{1}-\gamma +t_{2}e^{ik_{2}} & 0 & \ldots & \frac{g_{m}e^{-ik_{2}m}}{\sqrt{L}}\\ \ldots & \ldots & \ldots & \ldots & \ldots & \ldots \\ \frac{g_{n}e^{ik_{1}n}}{\sqrt{L}} & \frac{g_{m}e^{ik_{1}m}}{\sqrt{L}} & \frac{g_{n}e^{ik_{2}n}}{\sqrt{L}} & \frac{g_{m}e^{ik_{2}m}}{\sqrt{L}} & \ldots & 0 \end{array}\right),$$
where the system has 2L lattice sites and one site for impurity. This matrix [Hamiltonian $H_{AB}\left(k\right)$] is no longer a block diagonal matric, since the impurity couples to the SSH chain. Hence, the Hamiltonian cannot be written as the form of d·σ as usual. As is known to all, a pure nonreciprocal SSH model has a chiral symmetry. The spectrum of a chiral symmetric Hamiltonian is symmetric. For any state with energy *E*, there is a chiral symmetric partner with energy −E. While this model with A−B couplings does not preserve chiral symmetry. However, when $g_{n}=0$ ($g_{m}=0$ is similar), this system can still preserve a chiral symmetry $\Gamma^{-1}H_{AB}\left(k\right)\Gamma =-H_{AA}\left(k\right)$ with(10)$$\Gamma =\left(\begin{array}{cccccc} 1 & 0 & 0 & 0 & \ldots & 0\\ 0 & -1 & 0 & 0 & \ldots & 0\\ 0 & 0 & 1 & 0 & \ldots & 0\\ 0 & 0 & 0 & -1 & \ldots & 0\\ \ldots & \ldots & \ldots & \ldots & \ldots & \ldots \\ 0 & 0 & 0 & 0 & \ldots & 1 \end{array}\right).$$
For this system ($g_{n}=0$) with an odd number of site basis, the zero mode obviously always exists.

Physically speaking, for $g_{n}=0\;$ (or $g_{m}=0$), this is like a small atom coupled to a nonreciprocal SSH chain. Here, for the qubit coupled to a waveguide at one point, we call it “small atom”. In this case, except for the sublattice site *B* at the *m*-th unit cell coupled to the atom as an effective boundary, the other parts of the SSH chain are similar to a chain with a *A* site at both ends. For this kind of boundary condition, there is always a zero mode as usual. In a word, when $g_{n}$ (or $g_{m}$) goes from a finite value to zero, topological phase transitions will be bound to happen.

Next, consider a system ([Disp-formula FD3b-entropy-27-00078]) consisting of an impurity coupled to a nonreciprocal SSH chain via A−A coupling. In Figure 4, we show absolute value of the spectrum as a function of $t_{1}$ with the same parameters as used in Figure 3. Note that there is always a zero mode in the gap for the A−A coupling, and this result does not change with the coupling strength. The Hamiltonian of the system in momentum space via the Fourier transformation can be described by(11)$$\begin{array}{cc} H_{AA}\left(k\right)= & H_{\mathrm{SSH}}\left(k\right)+H_{I,AA}\left(k\right)\\ = & \sum_{k}\left[\left[\left(t_{1}+\gamma \right)+t_{2}e^{-ik}\right]{\hat{C}}_{A,k}^{†}{\hat{C}}_{B,k}+\left[\left(t_{1}-\gamma \right)+t_{2}e^{ik}\right]{\hat{C}}_{B,k}^{†}{\hat{C}}_{A,k}\right]\\ + & \frac{1}{\sqrt{L}}\sum_{k}\left[\left|e\rangle \langle g\right|{\hat{C}}_{A,k}\left(g_{n}e^{ikn}+g_{m}e^{ikm}\right)+H.c.\right]. \end{array}$$
Similarly, the matrix form of Hamiltonian ([Disp-formula FD3b-entropy-27-00078]) in the momentum space can be expressed as(12)$$H_{AA}\left(k\right)=\left(\begin{array}{cccccc} 0 & t_{1}+\gamma +t_{2}e^{-ik_{1}} & 0 & 0 & \ldots & \frac{g_{n}e^{-ik_{1}n}+g_{m}e^{-ik_{1}m}}{\sqrt{L}}\\ t_{1}-\gamma +t_{2}e^{ik_{1}} & 0 & 0 & 0 & \ldots & 0\\ 0 & 0 & 0 & t_{1}+\gamma +t_{2}e^{-ik_{2}} & \ldots & \frac{g_{n}e^{-ik_{2}n}+g_{m}e^{-ik_{2}m}}{\sqrt{L}}\\ 0 & 0 & t_{1}-\gamma +t_{2}e^{ik_{2}} & 0 & \ldots & 0\\ \ldots & \ldots & \ldots & \ldots & \ldots & \ldots \\ \frac{g_{n}e^{-ik_{1}n}+g_{m}e^{-ik_{1}m}}{\sqrt{L}} & 0 & \frac{g_{n}e^{-ik_{2}n}+g_{m}e^{-ik_{2}m}}{\sqrt{L}} & 0 & \ldots & 0 \end{array}\right).$$
The corresponding transcendental equation for energy *E* satisfies (See Appendix A for analytical results)(13)$$E=\frac{1}{L}\sum_{k}\left(\frac{g_{n}^{2}+g_{m}^{2}+2g_{n}g_{m}\cos\left[k\left(m-n\right)\right])}{E^{2}-\omega_{k}^{2}}\right).$$
Obviously, E=0 is always the solution of Equation (Equation 13), which can be seen in the numerical spectra of Hamiltonian $H_{AA}$ (Figure 3a–d). Fortunately, this system always has a chiral symmetry $\sigma^{-1}H_{AA}\left(k\right)\sigma =-H_{AA}\left(k\right)$ with(14)$$\sigma =\left(\begin{array}{cccccc} 1 & 0 & 0 & 0 & \ldots & 0\\ 0 & -1 & 0 & 0 & \ldots & 0\\ 0 & 0 & 1 & 0 & \ldots & 0\\ 0 & 0 & 0 & -1 & \ldots & 0\\ \ldots & \ldots & \ldots & \ldots & \ldots & \ldots \\ 0 & 0 & 0 & 0 & \ldots & -1 \end{array}\right).$$
Even after setting $g_{n}=0$ or $g_{m}=0$, we find that chiral symmetry will not vanish. This further indicates that zero mode will always exist.

### 3.2. Localization of Zero Mode

In the previous section, we mainly found that asymmetric couplings can cause topological phase transitions for A−B couplings. To illustrate this result again, we show the relationship between the localization of middle bound states and asymmetric couplings. To begin with, we define the population as modular square of the wave function $|\Psi_{A(B)}{|}^{2}$ in real space, and the populations of middle bound states for the system ([Disp-formula FD3a-entropy-27-00078]) with A−B coupling with different parameters $g_{n}$ are shown in Figure 5a–d. Here, the parameters are $t_{1}=1$, $g_{n}=0.5$, 0.3, 0.1, and 0.01 for (a), (b), (c), and (d), respectively. The other parameters are the same as used in Figure 3a–d. One can see that middle bound states have no obvious symmetry for spatial distribution when $g_{n}=0.5$, as shown in Figure 5a. However, the spatial distribution of the bound state gradually becomes an exponential decay (type of zero mode) as $g_{n}$ almost vanishes from a finite value, as shown in Figure 5d. This visually indicates that reducing one of the coupling strength can induce a zero mode.

Next, we are interested in analyzing localization of zero mode for the system wherein emitters couple to a nonreciprocal SSH chain. The populations of the zero mode as a function of site *N* with different parameters are shown in Figure 6 for A−B coupling. The population on the impurity was set to N=2L+1=101. The bars represent numerical results and the empty circles represent the analytical results. The corresponding coefficients of wave function are as follows (See Appendix B for analytical results):(15)$$\begin{array}{cc} \; & B_{l}/U_{e}=\left\{\begin{array}{cc} \frac{g_{n}}{t_{1}+\gamma }{\left(-\frac{t_{1}+\gamma }{t_{2}}\right)}^{\left(n-l\right)}, & (l<n),\\ 0, & (l\geq n), \end{array}\right.\\ \; & A_{l}/U_{e}=\left\{\begin{array}{cc} 0, & (l\leq m),\\ \frac{g_{m}}{t_{1}-\gamma }{\left(-\frac{t_{1}-\gamma }{t_{2}}\right)}^{\left(l-m\right)}, & (l>m), \end{array}\right. \end{array}$$
where $B_{l}$ ($A_{l}$) denotes the amplitude in the B(A) sublattice site of the *l*-th unit cell. Equation (Equation 15) shows that amplitudes in the wave function occupy B(A) lattice sites on the left (right) side of the impurity, and they satisfy exponential decay. Assuming l=n−1, we have $|B_{n-1}/U_{e}\left|=\right|g_{n}/t_{2}|$, and assuming l=m+1, we have $|A_{m+1}/U_{e}\left|=\right|g_{m}/t_{2}|$. This means that the zero modes mainly occupies *B* sites on the left side of the impurity when $g_{n}\gg g_{m}$ and occupies the *A* sites on the right side of the impurity when $g_{m}\gg g_{n}$. This has been clearly shown in Figure 6a–d. In addition, with the increase in $g_{n(m)}/t_{2}$, the populations of zero mode on the impurity are gradually suppressed. This means that the spatial distribution of zero modes can be regulated by coupling strengths.

For the system with A−A coupling, zero modes always exist regardless of the value of parameters. Simple algebra (see Appendix B for analytical results) shows that amplitudes of zero modes take $A_{l}=0$, and(16)$$B_{l}/U_{e}=\left\{\begin{array}{cc} \frac{g_{n}}{t_{1}+\gamma }{\left(\frac{-t_{1}-\gamma }{t_{2}}\right)}^{\left(n-l\right)}+\frac{g_{m}}{t_{1}+\gamma }{\left(\frac{-t_{1}-\gamma }{t_{2}}\right)}^{\left(m-l\right)}, & (l<n),\\ \frac{g_{m}}{t_{1}+\gamma }{\left(\frac{-t_{1}-\gamma }{t_{2}}\right)}^{\left(m-l\right)}, & (n\leq l<m),\\ 0, & (m\leq l), \end{array}\right.$$
with $-t_{2}+\gamma <t_{1}<t_{2}-\gamma$ and(17)$$B_{l}/U_{e}=\left\{\begin{array}{cc} 0, & (l<n),\\ \frac{-g_{n}}{t_{1}+\gamma }{\left(\frac{-t_{2}}{t_{1}+\gamma }\right)}^{l-n}, & (n\leq l<m),\\ \frac{-g_{n}}{t_{1}+\gamma }{\left(\frac{-t_{2}}{t_{1}+\gamma }\right)}^{l-n}+\frac{-g_{m}}{t_{1}+\gamma }{\left(\frac{-t_{2}}{t_{1}+\gamma }\right)}^{l-m}, & (m\leq l), \end{array}\right.$$
with $t_{1}>t_{2}+\gamma$ or $t_{1}<-t_{2}-\gamma$. For $-t_{2}-\gamma <t_{1}<-t_{2}+\gamma$ and $-t_{2}-\gamma <t_{1}<-t_{2}+\gamma$, the analytical results of the zero mode are not given. Similarly, the spatial distribution of the zero modes can also be changed by tuning the coupling strengths $g_{m}$ and $g_{n}$, as shown in Figure 7. It is clear that the analytical results (empty circles) given by Equation (Equation 16) are in good agreement with the numerical results (bars).

In this section, we mainly show that the localization of the zero modes is regulated by tuning the coupling strengths $g_{m}$ and $g_{n}$ for A−B couplings and A−A couplings. A notable feature is that the localization of the zero modes can be flexibly adjusted; as well, nonreciprocal hopping and asymmetric nonlocal couplings play a decisive role in systems. Concretely speaking, the impurity is nonlocally coupled to an SSH chain at two points and acts as two effective boundaries for the chain. Zero modes can localize at one of the two effective boundaries or both of them by tuning the coupling strengths $g_{m}$ and $g_{n}$. While for a nonreciprocal SSH chain, zero modes only localize at one of the two boundaries due to the non-Hermitian skin effect. For Hermitian systems, zero modes localize at both the two boundaries. It is obviously seen that the localization of the the zero modes of these two systems is relatively fixed. In short, we find a zero mode that can be regulated flexibly. In addition, except for the zero modes in the gap, there are other eigenvalues outside the continuous bands, as shown in Figure 3a. We call the corresponding eigenstates upper bound states. We find that these bound states can also be regulated by the coupling strengths $g_{m}$ and $g_{n}$. See Appendix C for the analytical results and numerical simulations.

### 3.3. Systems with Disorder

We further explore the localization of all eigenstates for systems in the A−A coupling case (A−B couplings are similar) with on-site disorder. To this end, a Hamiltonian with disorder reads as(18)$$H_{\mathrm{dis}}=H_{\mathrm{SSH}}^{\prime }+H_{I,AA},$$
with(19)$$\begin{array}{cc} H_{\mathrm{SSH}}^{\prime }= & \sum_{l=1}^{L}[\left(t_{1}+\gamma_{1}\right){\hat{C}}_{A,l}^{†}{\hat{C}}_{B,l}+\left(t_{1}-\gamma_{1}\right){\hat{C}}_{B,l}^{†}{\hat{C}}_{A,l}\\ \; & +t_{2}{\hat{C}}_{A,l+1}^{†}{\hat{C}}_{B,l}+t_{2}{\hat{C}}_{B,l}^{†}{\hat{C}}_{A,l+1}\\ \; & +V_{l}{\hat{C}}_{A,l}^{†}{\hat{C}}_{A,l}+V_{l}{\hat{C}}_{B,l}^{†}{\hat{C}}_{B,l}] \end{array}$$
where $V_{l}$ denotes an on-site disorder potential. Here, we set $V_{l}=SR_{l}$, where *S* is the disorder strength, and $R_{l}$ is a normal random number.

This localization behavior of all eigenstates can be easily quantified by the mean center of mass (mcom) of the amplitude squared of all eigenvstates $|\Psi_{R,n}\rangle$ averaged over many disorder realizations as follows:(20)$$\begin{array}{cc} \;\mathrm{mcom}\; & =\frac{\sum_{\ell =1}^{N-1}\ell {\langle A\left(\ell \right)\rangle }_{V}}{\sum_{\ell =1}^{N-1}{\langle A\left(\ell \right)\rangle }_{V}}, \end{array}$$
with(21)$$\begin{array}{c} {\langle A\left(\ell \right)\rangle }_{V}={\langle \frac{1}{N}\sum_{n=1}^{N}{\left|\langle \ell |\Psi_{R,n}\rangle \right|}^{2}\rangle }_{V}, \end{array}$$
where ${\langle \cdot \rangle }_{V}$ indicates disorder averaging, and we are only interested in the wave functions living at the SSH chain, so the population of the impurity is not considered.

As shown in Figure 8, we plot the mcom as a function of coupling strength $g_{n}$ and disorder strength *S* for the system with A−A coupling. The results were averaged for 50 disorder realizations. Note that one can clearly see that in the limit of small disorder strength *S* and small coupling strength $g_{n}$, all eigenstates localize at the right coupling point (white region in Figure 8). Thus, making the coupling strength $g_{n}$ larger will gradually increase localization of the eigenstates to the left coupling point (black region in Figure 8). In addition, the system exhibits a different localization behavior with the increase in the disorder strength *S*. On-site potential disorder will always dominate, leading all eigenstates to localize on the basis of Anderson localization (red region in Figure 8). The localization behavior is that eigenstates are randomly localized at points in the SSH chain.

## 4. Conclusions

In summary, we have studied an impurity coupled to a nonreciprocal SSH chain with asymmetric couplings. We show the fundamental properties of non-Hermitian spectra and find that asymmetric couplings can cause topological phase transitions for the A−B couplings case. In addition, the interplay of asymmetric couplings and nonreciprocal hopping can induce flexibly adjusted zero modes. They are localized at one of the two effective boundaries or both of them by tuning the coupling strengths $g_{m}$ and $g_{n}$. It is worth noting that the analytical results of the zero modes are precise and almost identify with the numerical results. We also explored the localization of all eigenstates for systems with on-site disorder, and we uncovered three types of localization behavior—localized at the right coupling point, localized at the left coupling point, and randomly localized at points in the SSH chain.

## Figures and Tables

**Figure 1 entropy-27-00078-f001:**
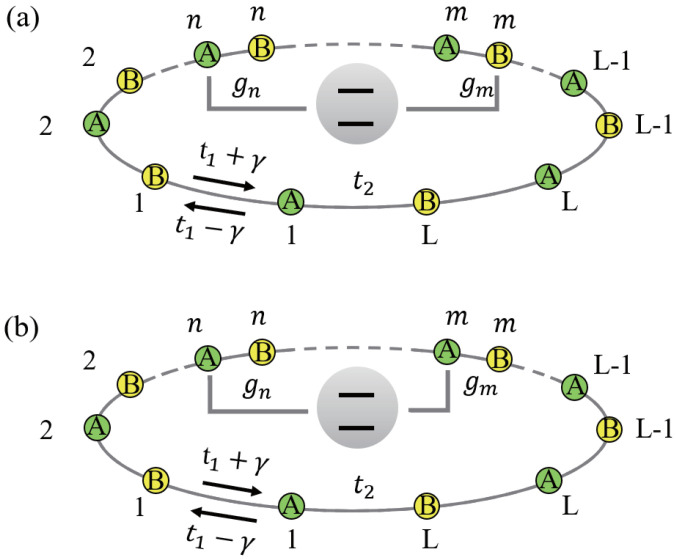
Schematicsof the nonreciprocal SSH chain coupled to an impurity via either A−B coupling (**a**) or A−A coupling (**b**) with asymmetric coupling strengths ($g_{m}\neq g_{n}$).

**Figure 2 entropy-27-00078-f002:**
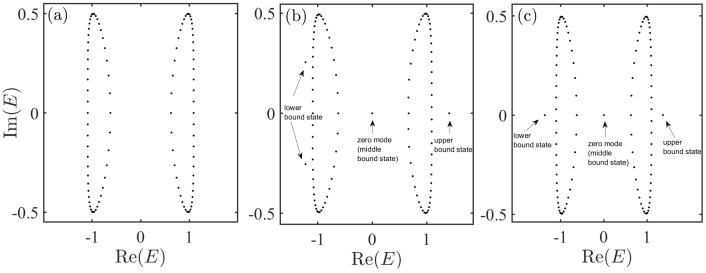
(**a**) Spectrum of pure SSH chain (Equation 1) in complex plane. (**b**) Spectrum of the system with A−B couplings ([Disp-formula FD3a-entropy-27-00078]) in complex plane. (**c**) Spectrum of the system with A−A couplings ([Disp-formula FD3b-entropy-27-00078]) in complex plane. The results were obtained by numerically solving the Schödinger equation. The parameters were set as $L=50,m=26,n=25,g_{n}=0.5,g_{m}=1,t_{1}=0.2,t_{2}=1,$ and γ=0.5.

**Figure 3 entropy-27-00078-f003:**
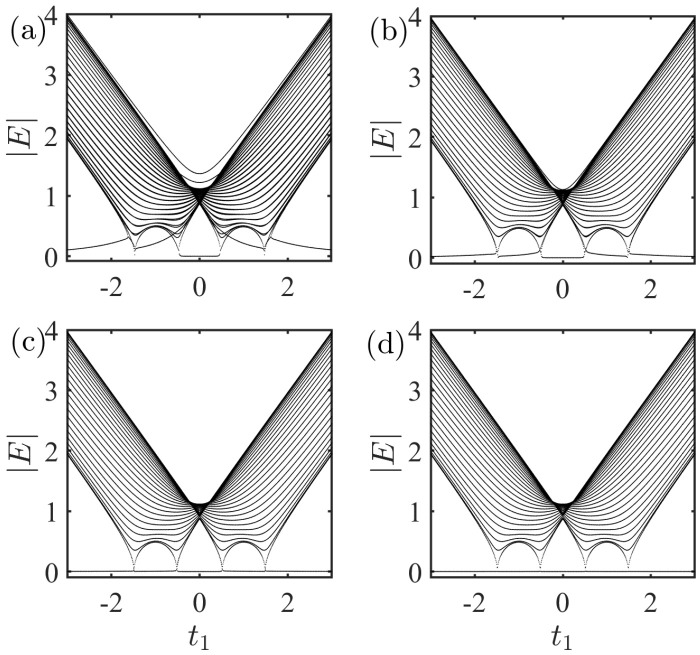
Absolute value of the spectrum as a function of $t_{1}$ with A−B coupling. The results are obtained by numerically solve the Schödinger equation. $g_{n}=0.5$, 0.1, 0.01, and 0.001 for (**a**), (**b**), (**c**), and (**d**), respectively. The other parameters were set as $L=50,m=26,n=25,g_{m}=1,t_{2}=1,$ and γ=0.5.

**Figure 4 entropy-27-00078-f004:**
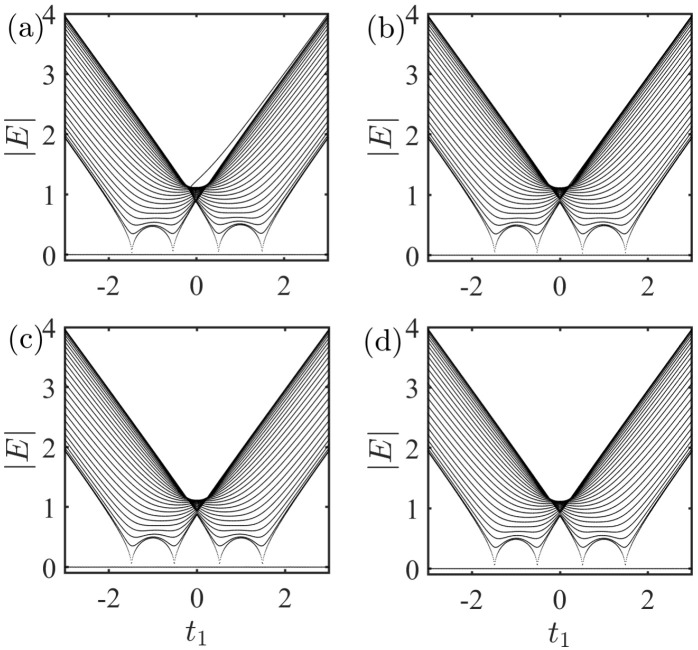
Absolute value of the spectrum as a function of $t_{1}$ with A−A coupling. The results were obtained by numerically solving the Schödinger equation. $g_{n}=0.5$, 0.1, 0.01, and 0.001 for (**a**), (**b**), (**c**), and (**d**), respectively. The other parameters were set as $L=50,m=26,n=25,g_{m}=1,t_{2}=1,$ and γ=0.5.

**Figure 5 entropy-27-00078-f005:**
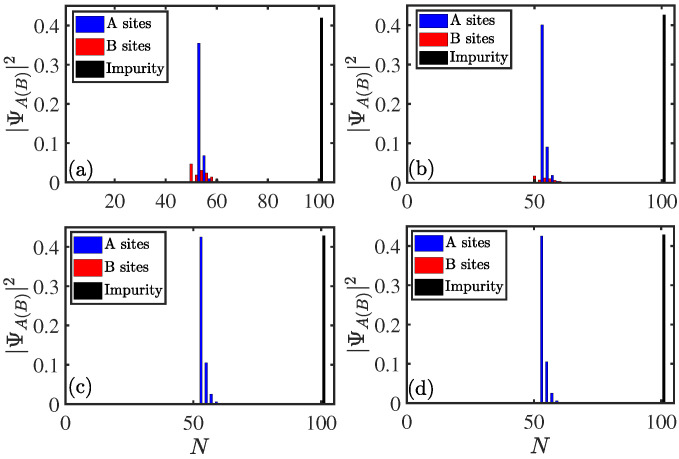
Populations of the middle bound states for the system ([Disp-formula FD3a-entropy-27-00078]) with A−B coupling as a function of site *N*. Here $g_{n}=0.5$, 0.3, 0.1, and 0.01 for (**a**), (**b**), (**c**), and (**d**), respectively. The other parameters were chosen as $L=50,m=26,n=25,g_{m}=1,t_{1}=1,t_{2}=1,$ and γ=0.5. The site of impurity was set to N=2L+1=101.

**Figure 6 entropy-27-00078-f006:**
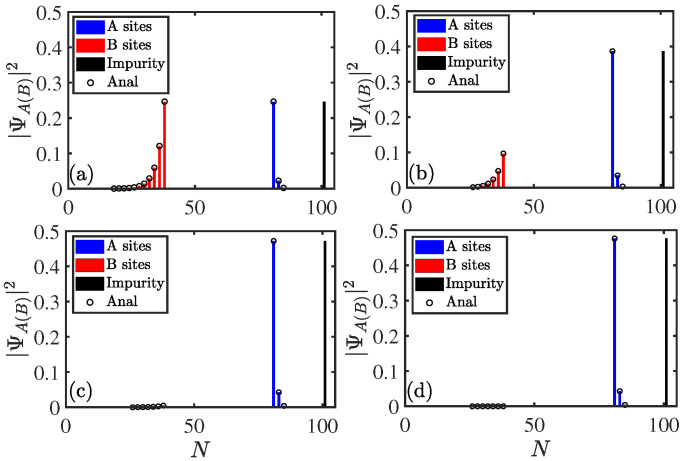
Populations of zero mode for the system ([Disp-formula FD3a-entropy-27-00078]) with A−B coupling as a function of site *N*. $g_{n}=1$, 0.5, 0.1, and 0.01 for (**a**), (**b**), (**c**), and (**d**), respectively. The other parameters were chosen as $L=50,m=40,n=20,g_{m}=1,t_{1}=0.2,t_{2}=1,$ and γ=0.5. The site of impurity was set to N=2L+1=101.

**Figure 7 entropy-27-00078-f007:**
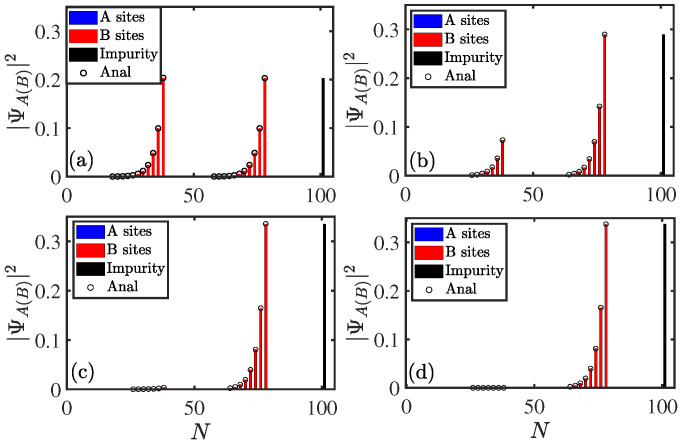
Population of zero mode for the system ([Disp-formula FD3a-entropy-27-00078]) with A−A coupling as a function of site *N*. $g_{n}=1$, 0.5, 0.1, and 0.01 for (**a**), (**b**), (**c**), and (**d**), respectively. The other parameters were chosen as $L=50,m=40,n=20,g_{m}=1,t_{1}=0.2,t_{2}=1,$ and γ=0.5. The site of impurity was set to N=2L+1=101.

**Figure 8 entropy-27-00078-f008:**
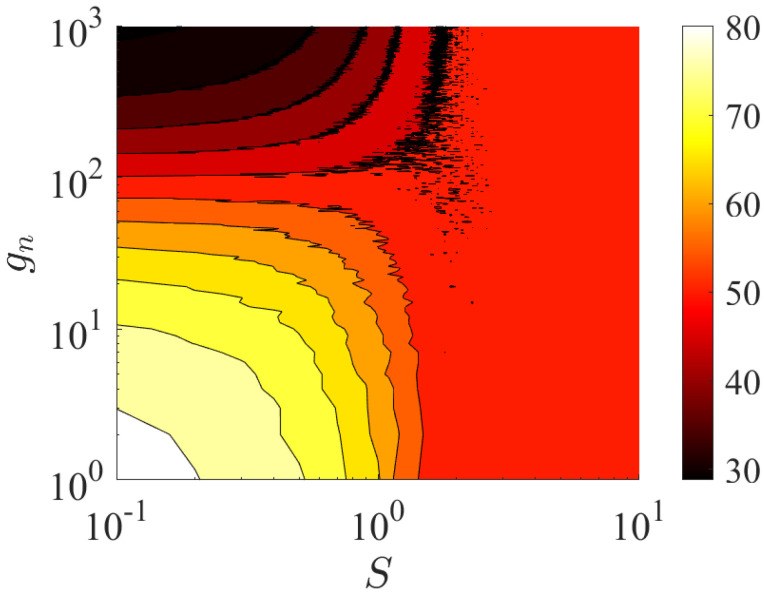
Mcom on the parameter space of coupling strengths $g_{n}$ and disorder strengths *S* for the system with A−A coupling. The results have been averaged for 50 disorder realizations. The parameters were chosen as $L=50,m=40,n=10,g_{m}=100,t_{1}=1,t_{2}=1,$ and γ=0.5. The site of impurity was set to N=2L+1=101.

## Data Availability

The data are contained within the article.

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
