# Peer review of "Anomalous Behavior of the Non-Hermitian Topological System with an Asymmetric Coupling Impurity"

_entropy, 2025, doi:10.3390/e27010078_

Round 1

Reviewer 1 Report

Comments and Suggestions for Authors

In the present paper the authors consider the problem of coupling a "giant atom" to a periodic SSH chain. This is a timely and interesting topic in view of recent interest in topology in non-hermitian systems and the associated bulk-boundary correspondence. 

However, I do not find the presentation very clear. I think the presentation would become much better if the authors in the main text present the Hamiltonian in momentum space, and not in the Appendix. (I find it very confusion that the lattice subspace indices A and B are called alpha and beta instead of A and B in momentum space. Why do the authors do this?) Then it is trivial to integrate out the SSH chain and get an effective Hamiltonian for the giant atom, and one can analyze the energy spectrum from that. In Figures 2 and 3, it would be nice to then also present the energy spectrum of the SSH chain, so we see more clearly what the coupling with the giant atom is doing. Also the figures would be much clearer by not plotting the absolute value of the energy, but just the energy, so that the gap is very clear now! In Figure 2 also bound states are denoted, but there is no discussion anywhere in the paper about these states. 

The notation for the wave function is also very confusing. Why do the authors not use Psi_A,i and Psi_B,i for instance, like on the creation and annihilation operators and so the probabilities are |Psi_A,i|^2 for instance like it is essentially common in all of quantum mechanics.

With such improvements in the presentation, I would recommend publication of this paper.  

Reviewer 2 Report

Comments and Suggestions for Authors

My comments are provided in the attached file.

Round 2

Reviewer 1 Report

Comments and Suggestions for Authors

The author has adequately responded to my previous referee report and the paper can now be published in its present form in my opinion. 

Reviewer 2 Report

Comments and Suggestions for Authors

The new title better corresponds to the content f the article. Plase correct: "an asymmetric coupling impurity."

The author took into account all the comments, made the necessary corrections and additions.